# Electrochemical Behaviour of Real-Time Sensor for Determination Mercury in Cosmetic Products Based on PANI/MWCNTs/AuNPs/ITO

**Noor Aini Bohari** [1], **Shafiquzzaman Siddiquee** [1,*], **Suryani Saallah** [1], **Mailin Misson** [1] **and Sazmal Effendi Arshad** [2]

1 Biotechnology Research Institute, Universiti Malaysia Sabah, Jalan UMS, 88400 Kota Kinabalu, Sabah, Malaysia; noorainibohari@gmail.com (N.A.B.); suryani@ums.edu.my (S.S.); mailin@ums.edu.my (M.M.)
2 Faculty of Science and Natural Resources, Universiti Malaysia Sabah, Jalan UMS, 88400 Kota Kinabalu, Sabah, Malaysia; sazmal@ums.edu.my
* Correspondence: shafiqpab@ums.edu.my

**Abstract:** Mercury is a common ingredient found in skin lightening soaps, creams, and makeup-cleansing products. It may cause skin rashes, skin discolouration, and scarring, as well as a reduction in the skin's resistance to bacterial and fungal infections. By looking at this scenario, developing a sensor that involved a simple procedure and fasts for real-time detection without affecting mercury sensitivity is urgently needed. For that reason, a fast and sensitive electrochemical method was developed to determine mercury in cosmetic products with the composition of polyaniline/multi-walled carbon nanotubes/gold nanoparticles/indium tin oxide sheet using methylene blue as a redox indicator. The significantly enhanced electrochemical performance was observed using cyclic voltammetry (CV) and differential pulse voltammetry (DPV). In order to detect mercury qualitatively and quantitatively, deposition potential and deposition time were respectively optimised to be 0.10 V and 70 s. The modified sensor was revealed a wide detection range of mercury from 0.01 to 10.00 ppm with a limit of detection of 0.08 ppm. The modified sensor towards mercury with a correlation coefficient ($r^2$) was of 0.9948. Multiple cycling, reproducibility, and consistency of different modified sensors were investigated to verify the modified sensor's performance. The developed sensing platform was highly selective toward mercury among the pool of possible interferents, and the stability of the developed sensor was ensured for at least 21 days after 10 repeated uses. The proposed method is a fast and simple procedure technique for analysing the mercury levels in cosmetic products.

**Keywords:** mercury sensor; mercury; cosmetic; cyclic voltammetry (CV); differential pulse voltammetry (DPV); electrochemical sensor

## 1. Introduction

The global cosmetics industry stood at $460 billion in 2014, according to the Global Cosmetics Market (2018), and is projecting to hit $675 billion by 2020 at an estimated annual growth rate of 6.4%. At CAGR (Compound Annual Growth Rate), the world cosmetics industry is projecting to rise by 4.3% from 2016 to 2022 [1]. This growing market calls for ongoing multidimensional surveillance (i.e., monitoring hazardous substances) and microbial pollution (i.e., chemical and biological pollution). Because of banned or limited substances under existing cosmetic laws, hazardous cosmetics pose a risk to customers and public health. According to the European Commission's Rapid Alert System (RAPEX), 62 cosmetic products were recalled, because of contamination by microorganisms from 2008 to 2014. Approximately 14 different countries notified the recalled products, with a higher number in 2013 and 2014 [2]. Therefore, monitoring the level of mercury ion (Hg2+)

in cosmetic products, environment and food products (recycled cooking oils) [3] are of fundamental importance.

The primary justification for using mercury is as a "Lightner" skin. It can act as a bleaching agent, and it has some preservative properties, which means it can help to lengthen the shelf-life of a product. High levels of mercury are used in cosmetics, mainly in products that promise dark spots, blemishes, and fine lines to disappear. Mercury is an efficient skin lightening substance that produces rapid results, but the price outweighs the benefits. Mercury is also used in many drug products as a preservative antifungal and antiseptic agents, such as some vaccines, ointments, contact lens solutions, and solutions for ear, eye, and nasal drop. It produced free radicals and categorised as having a carcinogenic effect on human health. Regular skin lightening products containing mercury can lead to skin rashes like dermatitis, eczema, acne venenata, keratosis [4], discolouration, and scarring, as well as a reduction in the skin's resistance to bacterial and fungal infections and even skin cancer [5]. Mercury is a highly volatile component with an extended atmospheric half-life, which leads to a more significant problem through accumulation, and its ubiquity in cosmetics may pose a significant health concern. Long-term exposure to high mercury levels in cosmetic products may cause serious health consequences, including damage to the kidneys and gastrointestinal and neurological disability. It can further transform microbial methylation into more poisonous methylmercury, accumulate in human bodies with a high enrichment factor, and then affect the nervous, immune, and digestive systems and cause severe damage to the kidney, liver and brain [6,7]. Other effects include anxiety, depression or psychosis, and peripheral neuropathy. Although the manufacture of some mercury-containing creams in America and some European nations are illegal, the use of mercury-based skin lightening cosmetics has increased worldwide.

The mercury toxicity symptoms, also known as the "hatters disease" as immortalised through Lewis Carroll in Alice in Wonderland, consist of psychological (recent memory disorder, mental function dysfunction, inattention, and depression) and neurological (irritability, memory loss, and neuropathy) problems [8]. Many adverse reactions reported with mercury toxicity include renal dysfunction (minimum change or nephritis of the membranes) [9,10]. Moreover, paradoxically, it has improved skin pigmentation. The latter occurs either through a rise in melanin production (mechanism unknown) or through direct deposition of metallic mercury granules in the dermis. Percutaneous mercury absorption occurs exclusively by cutaneous appendages, and mercury deposition perifollicular accentuation in mercury-induced hyperpigmentation is thus detected by lesion biopsy [11]. Interestingly, pregnant or lactating women's use of mercury agents for skin lightening has also been related to their neonates' adverse effects, including the possibility of anaemia, renal dysfunction, and cataracts. It has produced widespread mercury-induced health effects in some cosmetics that may cross the placenta to meet the foetus; pregnant women and women of childbearing age should avoid mercury-based cosmetics.

In China, reports of adverse cases have raised public concern with ever-increasing cosmetic use. In an investigation in a Hong Kong community, the mean urinary mercury concentrations of 286 users of one cosmetic product were 45.2 µg/L, and 65% of participants had elevated urine mercury concentrations. Concentrations of urinary mercury were significantly higher among people who last used the cream within 45 days than others who did not. Studies have shown that women who use mercury-containing creams in their urine reached mercury levels from 0.03 to 0.15 mg/L [12]. At these concentrations, the central nervous system and the kidneys are at significant risk of adverse effects. Accordingly, Sun et al. [13], from 2009 to 2017, the current research included 16 female patients at the First Associated Hospital of Zhengzhou University diagnosed with chronic mercury intoxication. These patients ranged in age from 19 to 50. Both 16 patients had flexible therapy, gradual onset, and a secret history that led to the delayed diagnosis. There was no history of mercury exposure in their work, but they all had a history of using skin-lightening products, which had immediate and drastic whitening effects. The chief symptom was miserable pain, and nonsteroidal anti-inflammatory medications, anti-epileptic drugs,

and serotonin-norepinephrine reuptake inhibitors were poorly responsible for limb, brain, abdominal, or lumbosacral pain. As 0 referred to no pain and 10 referred to baryodynia, according to the World Health Organization visual analogue scale rating. Between 4 and 7, the pain of these 16 patients was rated. Another feature was renal injury, in which six of all the patients had proteinuria. Moreover, there were varying degrees of irritability, insomnia, nightmares, depression, anxiety, and memory loss in all patients. Also, eyelid, tongue, or limb tremors in five patients, while gingivitis was present in three patients [13]. Mercury ions replace tyrosinase enzyme anions, which inhibit melanin formation and create the effects of whitening and anti-freckle [1]. To make the lightening effect exceptional, certain items are added with the mercury material, which exceeds bid badly in thousands or even millions of times. Chronic mercury poisoning could be caused by the long-term use of these unqualified goods. Irritability, tremors, and gingivitis are the typical signs of chronic mercury poisoning, although the onset of pain is uncommon. Also, memory loss, dizziness, insomnia, dreams, oedema, proteinuria, stomach pain, nausea, hyperthyroidism, and abortion have been reported to cause it. There are unusual reports on mercury toxicity, which leads to damage to peripheral nerves. Patients whose reaction, seizures, confusion, trouble swallowing, glossolalia, deafness, narrow field of vision, strabismus, or photophobia were prevalent in their study like Minamata disease. Mercury affects the nerves in many ways: it interferes with the roles of the membrane and receptor and the transport and metabolism of neurotransmitters; it interrupts cytoskeletal proteins and interferes with axoplasmic movement and signal transduction; it interferes with cell respiration, metabolism of energy and others. In Arizona, a 17-month-old infant suffered hypertension, fussiness, constipation, and arthralgia from a skin-lightening cosmetic used by her mother and grandmother due to mercury toxicity. The blood mercury level was 26 mcg/L, and creatinine was 243 mcg/g in the urine. She was sucimer-chelated, home polluted, and remediation is required [14]. It is unclear how effective mercury-containing products are as skin-bleaching agents. No well-controlled studies have been given to the Food and Drug Administration to document those products' efficacy. While mercurial preservatives are recognised as highly effective, there are less toxic and adequate alternatives available, except for certain cosmetics in the eye area.

The United States Food and Drugs Administration (US FDA) set a value of less than 1 ppm for mercury in cosmetics and its derivatives in 1992, due to mercury toxicity. Many studies were carried out to identify and quantify mercury in cosmetic preparations and detected too high mercury levels [15]. In Europe and China, the cap for cosmetic products for eyes and lips is 0.07 ppm. Likewise, Japan strictly prohibits the use of mercury in untraceable cosmetics [1]. Table 1 is permitted the summary of limits in cosmetics products.

**Table 1.** Summary of Regulations Regarding Mercury Concentration in Cosmetics.

| Regulatory Body | Limits for Cosmetics Other than Eye Area Products |
| --- | --- |
| European Union | Banned |
| Many African Nations | Banned |
| Japan | Banned |
| The United States and Drug Administration | <1 ppm |
| Health Canada | $\leq$3 ppm |
| Philippines Food and Drug Administration | $\leq$1 ppm |
| **Regulatory Body** | **Limits for eye area products** |
| European Union | $\leq$0.007% by weight |
| The United States Food and Drug Administration | $\leq$65 ppm expressed as mercury (approximately 100 mg/kg expresses as phenylmercuric acetate or nitrate) |

The conventional analytical methods for detection of $Hg^{2+}$ in cosmetic products usually rely on the use of advanced instruments based on spectrometric and chromatographic

techniques such as atomic absorption spectrometry (AAS) [6,16,17], graphite furnace atomic absorption spectrometry (GFAAS) [7], inductively coupled plasma atomic emission spectrometry (ICP-AES) [18], inductively coupled plasma mass spectrometry (ICP-MS) [19] and liquid chromatography inductively coupled plasma mass spectrometry (LC-ICP-MS) [20], high-performance liquid chromatography (HPLC) with ICP-MS [20], inductively coupled plasma atomic emission spectrometry (ICPAES) [18] and cold vapor atomic absorption spectrometry (CVAAS) [21–23] direct mercury analyser [7,24]. Such approaches are proven to be highly accurate and sensitive, complicated, expensive, and time-consuming procedures are the significant downsides for daily analysis and on-site detection of many samples in the field. Alternatively, several techniques for $Hg^{2+}$ ion detection were developed, such as colourimetric [25,26], electrochemical [27–29], and surface-enhanced Raman scattering [30–32]. While such approaches are proven to be highly accurate and sensitive, the complex, expensive, and time-consuming procedures associated with these methods called for a simpler alternative for daily analysis and on-site detection of a large number of samples in the field. Despite these advances, a novel sensor with simple operation, low cost, quick response, high sensitivity, selectivity, and reliability for real-time on-site monitoring to perform its identification and quantification are still required. Sensors are one of the alternatives with three main characteristics: the element of recognition (enzyme, antibody, DNA, etc.), the structure of signal transduction (electric, optical or thermal), and the element of amplification as in Figure 1.

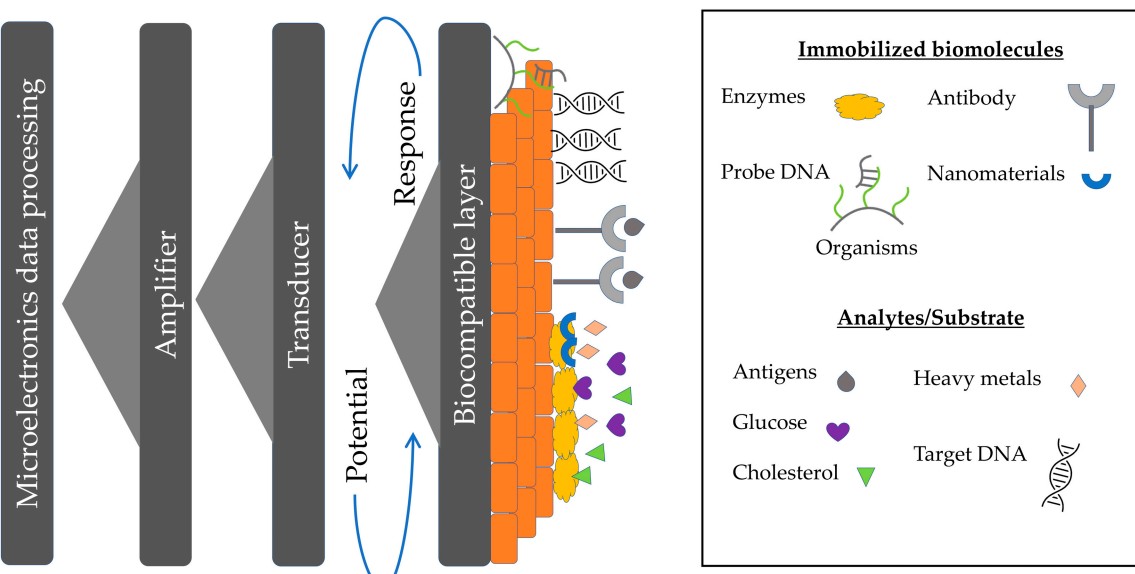

**Figure 1.** The schematic diagram of a biosensor incorporates a living organism or product derived from a living system as the recognition element or a bioreceptor and a transducer to convert a biological reaction into a measurable signal or indication.

The current analytical techniques were laborious, highly time-consuming, and incapable of on-site detection. The electrochemical sensor has been an ideal tool for detecting heavy metal ions, including $Hg^{2+}$, attributed to its simplicity, quick response, low-cost, and portability. In electrochemical sensing, the type of working electrode is among the key factors that influenced the analytical performance. Metal nanoparticles and carbon nanotubes have usually been used as working electrodes due to their large surface area and excellent structural, electronic, and optical properties [33]. The major limitation of an electrode made from precious metals is the high cost and complicated manufacturing. The carbon-based electrode is cheap and easy to prepare but lacks stability and poor reproducibility. The bare ITO sensitivity for such an application, however, was not costly compared to others. In this regard, modification of the ITO film is necessary. This study expected to enhance the

current technique by involving simple procedures and fast detection for real-time on-site monitoring without affecting the detection's sensitivity.

ITO film is of great interest as a cheap transparent electrode in electrochemical sensors' production because of its high optical visibility, strong electrical conductivity, and stable surface properties [34–36]. The bare ITO sensitivity for such an application, however, was not high. In this regard, modification of the ITO film is necessary. Carbon-based materials such as multi-walled carbon nanotube (MWCNTs), graphene oxide, and precious metals such as gold and silver have been widely used to develop electrodes for the electrochemical sensor. MWCNTs are often used in biosensors' development due to their exceptional properties, including high aspect ratio, excellent electrical conductivity, and exceptionally high mechanical strength and stiffness [37]. The mixture of MWCNTs and polymers not only possesses the properties of each part with synergistic effects but is also of significance due to its ease design and ability to integrate the elements of biorecognition into its porous structures [38]. The dispersion of MWCNTs into the PANI matrix for the manufacture of MWCNTs/PANI composites showed significant electrical conductivity. Polyaniline (PANI) is preferred among the conductive materials to prepare flexible transparent electrodes due to its ease of synthesis, high conductivity, and environmental stability. It serves as an interconnected nanostructured scaffold for the homogeneous distribution of AuNPs and synergistically enhanced response by the resulting PANI/AuNPs nanocomposite [39]. Many researchers reported that conductive composite films blend PANI with other polymers or fillers, thus producing materials with improved conductivity and processing capabilities [40,41]. Gold nanoparticles (AuNPs) are increasingly used in many electrochemical applications because they could enhance electrode conductivity and facilitate electron transfer, thus improving analytical selectivity and sensitivity. Based on their nano dimensional properties and favourable electrocatalytic behaviour, we were developed an electrochemical sensor based on multi-walled carbon nanotubes (MWCNTs) and gold nanoparticles (AuNPs) that could increase sensitivity and electroanalytical signals.

## 2. Materials and Methods

### 2.1. Reagents and Chemicals

All the reagents were of analytical grade and were used without further purifications. Sigma-Aldrich purchased Tris- (hydroxymethyl) aminomethane-HCl (Tris-HCl), MWCNTs, Methylene blue (MB), gold nanoparticles (AuNPs), polyaniline (PANI), ITO, and standard mercury solution. All solutions and subsequent dilutions were prepared using deionised water from a filter instrument model, namely milli-Q (Millipark ®40), and carried out at laboratory temperature conditions of $23.0 \pm 2.0\,°C$.

### 2.2. Instrumentations

Electrochemical cyclic voltammetry (CV) and differential pulse voltammetry (DPV) measurements were recorded using an μ Autolab PGSTAT 30 computer-controlled potentiostat (EcoChemie, Netherlands) with a standard three-electrode system. The modified ITO sheet-coated glass (Sigma-Aldrich, 60 Ω resistance) electrode (working area $1 \times 1$ cm$^2$) served as a working electrode, while platinum wire (Pt) was used as a counter electrode with an Ag | AgCl | KCl 3 M reference electrode completing the cell assembly as in Supplementary Figure S1. All experiments were conducted at a room temperature condition of $23.0 \pm 2.0\,°C$—the conceptual experiment. Statistical analysis of the CV and DPV responses were performed using SPSS14.0 (SPSS, Inc., Chicago, IL, USA). Results were conducted by the one-way analysis of variance followed by Tukey's post hoc test. All data were presented as mean $\pm$ STD. The criterion for statistical significance was $p < 0.05$. The experiments were repeated five times ($n = 5$) independently.

### 2.3. Electrode Preparation and Modification

The ITO electrode modification was performed using the electrochemical deposition technique following the method described by Bohari et al. [42]. Briefly, the ITO flexible sheet

was manually peeled off, followed by 1 M HCl's electrodeposition in aniline solution onto the ITO film. A 1 mg of MWCNTs was suspended in a 1 mL concentrated mixture of $H_2SO_4$ and $HNO_3$ in 3:1 ration (v/v) and subjected to ultrasonication for 2 h in order to obtain a homogeneously distributed black solution. The functionalised MWCNTs were washed thoroughly with distilled water to extract the acid. PANI-coated ITO electrode was dipped for 24 h in the MWCNT-COOH and AuNPs. The resulting PANI/MWCNTs/AuNPs-modified ITO electrode was thoroughly washed with distilled water to remove unbound materials and kept at 4 °C in a dry petri dish. The modified ITO electrode's surface was soaked in 1 mM of MB for 2 min after effective immobilisation, followed by washing with Tris-HCl buffer (pH 7.0) to eliminate unspecific physical adsorption.

### 2.4. Experimental Setup

All electrochemical characterisation and measurement were carried out using a conventional three-electrode system with a modified ITO electrode as the working electrode, a Pt wire as the auxiliary electrode, and Ag | AgCl (saturated 3 M KCl) electrode as the reference electrode. In a three-electrode cell containing 11 mL of 0.05 M at pH 7.0 Tris-HCl buffer containing 1 mL of 6 ppm standard mercury, CV measurements were carried out. Current measurements were performed in the potential range between +0.0 and +2.0 V using 0.1 Vs$^{-1}$, in a 50 mM Tris-HCl buffer (pH 7.0) in the presence of Methylene Blue (MB) as a redox indicator. All electroanalytical measurements were conducted at room temperature. All measurements were carried out at the Biochemistry laboratory using deionised water to prepare reagents and clean in room temperature conditions of 23.0 ± 2.0 °C.

## 3. Results and Discussion

MWCNTs N-alkyl chain length is due to Van der Waals interactions between nanoparticles (AuNPs) and nanocomposite (MWCNTs). Nanoparticles are then attached to MWCNTs and PANI network matrix and have increased in surface area to improve mercury detection sensitivity. PANI can dissolve many transition metal complexes; it increases electrochemical sensor reaction rates and selectivity [43]. Regularly, the increasing oxidation peak currents indicated that the combination of MWCNTs/AuNPs attached to the PANI nanocomposite became highly conductive materials that persistently deposited on the ITO sheet. Hence, PANI/MWCNTs/AuNPs effectively immobilised onto the ITO surface in MB presence and provided the necessary conduction pathways to determine mercury. This phenomenon indicated a strong electrostatic repulsion between PANI/MWCNTs/AuNPs, and the negatively charged ITO electrode [44,45]. The solution without modification was used as a negative control. The sensor's conductivity efficiency comprising PANI/MWCNTs/AuNPs/ITO was significantly enhanced in the potential current due to the combination of nanoparticles and nanomaterials.

The synergistic action between high conductivity and stability of PANI, high aspect ratio and exceptional mechanical strength of MWCNTs, and favourable electrocatalytic behaviour of AuNPs could significantly enhance the sensor sensitivity and electroanalytical signals. In this perspective, herein, PANI/MWCNTs/AuNPs-modified ITO electrode was developed by direct electrodeposition of PANI, MWCNTs, and AuNPs on the ITO sheet. The interactions between PANI and carboxyl functional MWCNTs take place due to electrostatic interaction between -COO- groups of MWCNTs and -NH+ of PANI, the hydrogen bonding between -OH groups on MWCNTs and -NH group of PANI, and also a small number of $-\pi$ staking between the $\pi$-bonded of the MWCNTs and the quinoid ring of aniline monomer [45]. Such strong interaction ensures that PANI is absorbed on the surface of MWCNTs, which serve as the core and the self-assembly template (deposition) during the formation of tubular nanocomposites. When the PANI is mixed with AuNPs, PANI's oxidation coincides, leading to in situ polymerised cause by strong interaction between MWCNTs and PANI.

### 3.1. Precision Test

The optimisations and characterisations of the PANI/MWCNTs/AuNPs/ITO electrode were followed by Bohari et al. [42]. In this study, the DPV calculation was explicitly used to analyse electrochemical behaviour. DPV is a sensitive technique that allows determining the analyte with high accuracy results in a short period. Multiple cycling, reproducibility and repeatability, storage stability, interference study, and spike recovery studies were conducted to verify the developed sensor's performances. Table 2 shows a summary of the electrochemical behaviour of the sensor in this study.

**Table 2.** Summary of the Behaviour Analysis of the Developed Electrode.

| Characteristic | Mean $\pm$ STD | RSD (%) |
|:---:|:---:|:---:|
| Multiple cycling | $1.52 \times 10^{-4} \pm 1.81 \times 10^{-6}$ | 1.93 |
| Reproducibility | $1.69 \times 10^{-4} \pm 2.31 \times 10^{-6}$ | 2.82 |
| Repeatability | $1.48 \times 10^{-4} \pm 3.54 \times 10^{-6}$ | 1.24 |
| Storage stability | 21 days, $95-99\%$ | |
| Interference study | $p < 0.05$, the hypothesis is accepted | |
| LOD | 0.03 ppm | |
| Spike recovery | $96.7-97.8\%$ | |

### 3.1.1. Limit of Detection (LOD)

Sensitivity is the capability to respond to changes in the concentration of analyte efficiently and measurably. Due to the high relative sensitivity compared to CV analysis, the DPV technique was used to measure mercury. The detection limit is the smallest quantity that differs significantly from the blank. It is observable when the signal is more significant than three times the noise but cannot be quantified accurately. The DPV of PANI/MWCNTS/AuNPs/ITO electrode in the presence of different standard mercury concentrations are shown in Figure 2. As the concentration of mercury increased, the anodic current increased. The inset shows the calibration curve of different mercury concentrations in 50 mM Tris-HCl (pH 7.0) with 1 mM MB as a redox indicator with a potential range of +0.4 to +2.0 V using 0.10 $\text{Vs}^{-1}$ scan rate according to Bohari et al. [42]. With increasing concentration of mercury, the peak currents increased linearly. The increment of mercury concentration attaches on the electrode surface, the peak potential shifted positively, and the separation of peak potential increased. Additionally, both anodic and cathodic peak currents grew.

The linear regression equation with a correlation value of $r^2 = 0.9948$ can be expressed as y = 1.174x − 1.3327. The mean value obtained from independent measurements corresponded to each point of the calibration graph. According to the equation below, a detection limit (LOD) of 0.08 ppm was obtained Equation (1). These results are attributed to the good adsorption capacity, the large specific surface area of the electrode, good interaction of mercury on the electrode surface, enhanced mercury electron transfers with the modified ITO, and excellent nanocomposite PANI/MWCNTs/AuNPs/ITO. More importantly, the ITO modification has enabled the detection of trace amounts of mercury with concentration from 10.00 to 0.01 ppm, suggesting high sensitivity of the method compared to other methods reported in Table 5.

$$\text{LOD} = 3\frac{\text{s}}{m} \tag{1}$$

s: Standard deviation of the blank measurements
*m*: Slope of the calibration curve
A: Area of the electrode

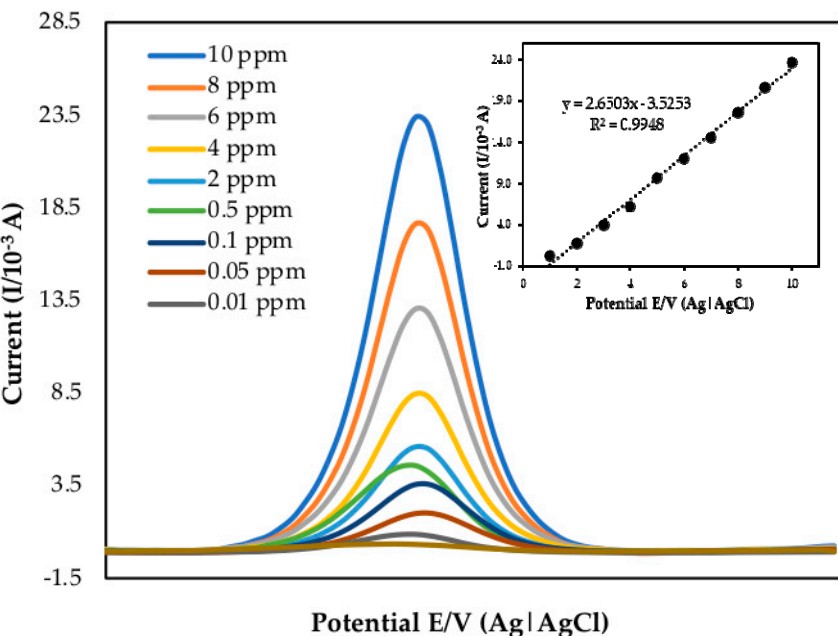

**Figure 2.** DPV on the different mercury concentrations with PANI/MWCNTS/AuNPs/ITO (pH 7.0 Tris–HCl, 1 mM MB, n = 5). Inset: Calibration curve between the oxidation peak currents obtained by DPV and mercury concentrations.

The mercury is attached to PANI/MWCNTs/AuNPS/ITO electrode via electrostatic interaction between the amino groups PANI and the carboxylic acid groups of functionalised MWCNTs together with the π-π interactions and hydrogen bonding due to the mercury complementary unit. As reported by Wang et al. [45], the mercury atoms and amino groups may form hydrogen bonds with the carboxyl group and amino group. Some researchers reported adding PANI to the modified electrode resulting in load transfer resistance decreasing and load transfer rate increasing due to higher PANI conductivity [46,47]. The PANI/MWCNTs/AuNPS/ITO showed a much lower determination limit, and broader linear range of mercury concentrations than other previously reported modified electrodes, which many attribute to the synergistic effect of the PANI/MWCNTs/AuNPs/ITO [48].

### 3.1.2. Multiple Cycling, Reproducibility and Repeatability Test

Multiple cycling, reproducibility, and repeatability studies were conducted to observe the occurrence of redox reaction and test the developed sensor (pH 7.0, Tris-HCl buffer, 50 mM Tris-HCl). The multiple cycling or electrode consistency is when the same quantity of one sample is repeated over the same electrode several times. It is a measure of electrode response variations. In this study, the multiple cycling studies were performed by setting 10 cycle numbers, and the potentiostat automatically generated the data with the time frame. Bohari et al. [42] reported that repeatability described the minimum, and reproducibility described the maximum variability in results. Reproducibility is interlaboratory accuracy when aliquots of the same sample are tested with the same modifications onto different electrodes. It was evaluated with the same mercury concentration on five consecutive days with five modified electrodes independently prepared in the same experimental conditions (pH 7.0, Tris-HCl, 1 mM MB). Repeatability is defined as the degree of single-coated sensor used continuously under similar operating conditions for a series of measurements that can produce the same result using the same substrate concentration [49,50]. Five successive DPV scans on the modified electrode's surface in 50 mM Tris-HCl (pH 7.0, 1 mM MB) were reported to investigate the modified electrode's repeatability. These results indicated that the modified ITO electrode possesses high precision since it has good reproducibility and acceptable multiple cycling and repeatability. Reproducibility studies were conducted on the same condition except on different days, and for 21 days, the findings were showed to

be less than 10% RSD [6]. Based on the results obtained (Figure 3), the peaks were close to each other and showed that the modified ITO provided high sensitivity and accuracy for mercury determination due to the constant of the current responses supported by Kulikova et al. [48] and Kuzin et al. [51]. These results indicated good reproducibility of the modified electrode.

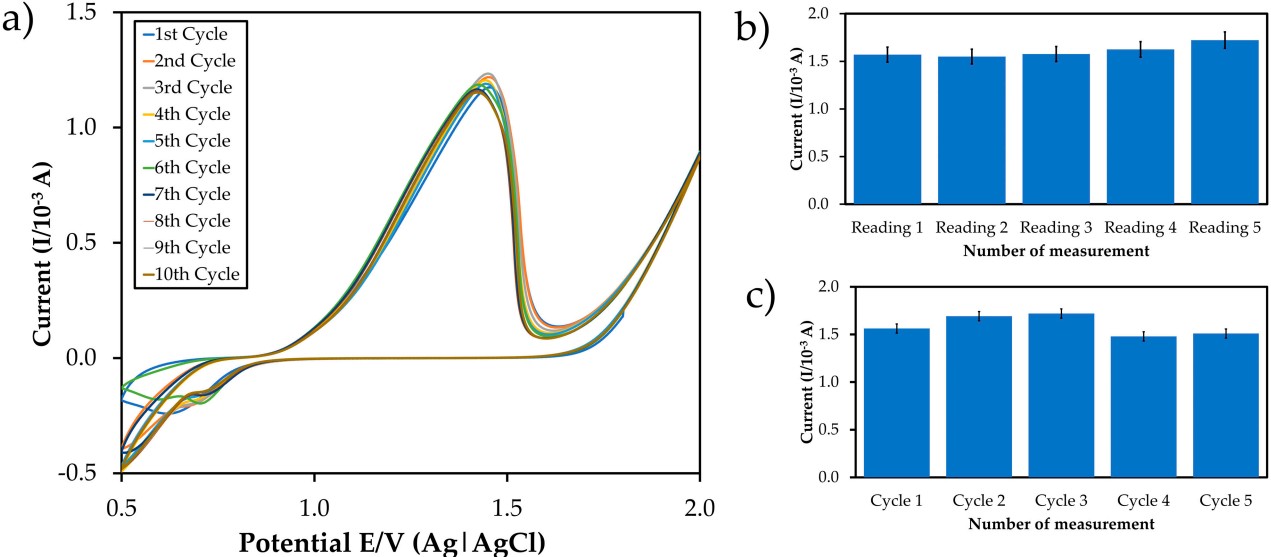

**Figure 3.** The cyclic voltammogram and the bar chart of (**a**) multiple cycling, (**b**) reproducibility, and (**c**) repeatability test of the modified ITO in the presence of standard mercury, in 50 mM Tris-HCl, pH 7.0, 1 mM MB, n = 5.

### 3.1.3. Storage Stability Test

The sensor's storage stability was analysed by periodically checking their current responses up to twelve (12) weeks, or 84th day. Figure 4 summarises the storage stability of the modified results of PANI/MWCNTs/AuNPs/ITO electrodes. The possible peak current response was slightly higher when the change was held for up to 14th day. After day 14, peak mercury currents decreased compared with the initially produced electrode to 4.1%. The results showed that the electrode response to mercury after 0, 7th, and 14th day was respectively 99, 98, and 95% of its original value. The slight decrease in the current response was indicated the long-term and high stability of the developed sensor for the determination of mercury. The PANI/MWCNTs/AuNPs/ITO electrode tends to have sufficient stability. A gradual decrease of reduction peaks was observed, indicating the chemical reaction's behaviour at the interphase of electrode dropped. No further interaction at the interphase of the modified electrode occurred efficiently with the analyte in the electrochemical cell. The free gold did not embed in MWCNTs to retain its movement, making it floating freely or suspended in electrochemical electrolyte might be contributed.

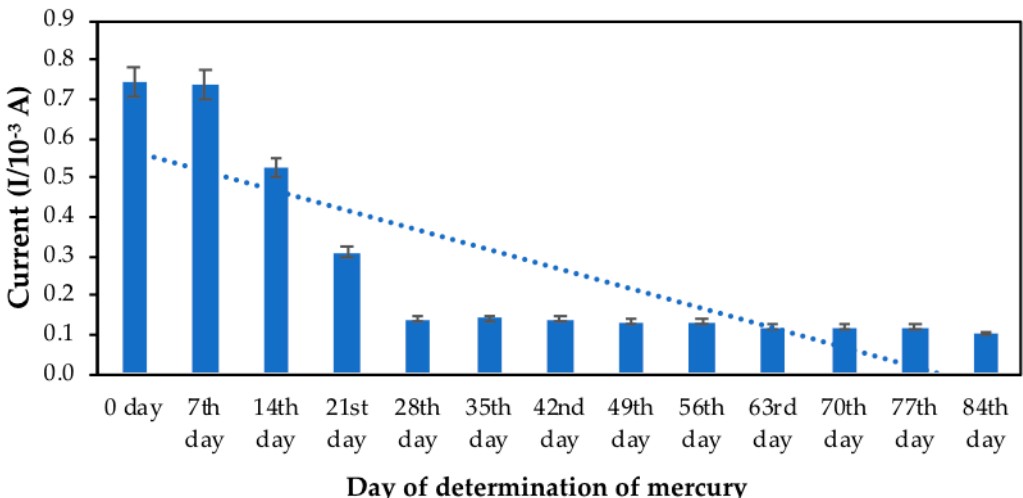

**Figure 4.** Bar chart of the developed sensor's storage stability in the presence of standard mercury, in 50 mM Tris-HCl, pH 7.0, 1 mM MB, n = 5.

### 3.1.4. Interference Studies

A series of experiments were conducted to assess the validated electrochemical method's interference further to verify the developed sensor's real-world applications. In the presence of certain compounds commonly found in cosmetic products such as pyrophosphate, papain, oligosaccharides, vitamin C, mannitol, collagen, amino acid, stearic acid, benzene, toluene, cetyl palmitate, methylene glycol, sodium chloride, potassium cetyl sulfate, and tea tree oil [52,53], the current responses were analysed to verify the selectivity of the sensor created. Interferences study can differentiate analytes from other molecules in the sample is selectivity (or specificity). Avoiding interference is selectivity.

Table 3 shows the selectivity of mercury and common disrupting compounds in cosmetics with the analytical buffer at the same mercury concentration using one-way analysis of variance (one-way ANOVA) followed by Tukey's post hoc statistical analysis ($p < 0.05$). ANOVA test tells whether there had an overall difference between the groups, but it did not mention which specific groups differed from Tukey Post Hoc test study. Tukey Post Hoc analysis was confirmed where the differences have occurred between groups when an overall statistically significant difference in group means (i.e., statistically significant one-way ANOVA result). The test compares all possible pairs of means. The results showed that the current mercury responses were significantly higher than other substances, suggesting that the sensor exhibited high mercury selectivity towards mercury than other compounds. The developed sensor's selectivity is satisfactory, and the PANI/MWCNTs/AuNPs/ITO capability is detected as specific target mercury molecules. The weak DPV signals were deserved in the absence of mercury despite the substitutions' high concentration. In contrast, in the presence of mercury, the DPV signal was high. The results indicated that the developed electrochemical sensor's selectivity was strong, allowing accurate mercury detection without other ion interference [35].

**Table 3.** One Way ANOVA and Tukey Post Hoc Test for Interference Study.

| | ANOVA Current | | | | |
|---|---|---|---|---|---|
| | Sum of Squares | df | Mean Square | F | Sig. |
| Between Groups | 20.918 | 7 | 2.988 | 2,038,564.012 | 0.000 |
| Within Groups | 0.000 | 32 | 0.000 | | |
| Total | 20.918 | 39 | | | |

| Post Hoc Tests | | | |
|---|---|---|---|
| (I) Selectivity | (J) Selectivity | Mean Difference (I–J) | Sig. |
| Mercury standard | Pyrophosphate | 0.0000031935 * | 0.000 |
| | Papain | 0.0000032770 * | 0.000 |
| | Oligosaccharides | 0.0000030656 * | 0.000 |
| | Vitamin C | 0.0000033192 * | 0.000 |
| | Mannitol | 0.0000029980 * | 0.000 |
| | Collagen | 0.0000032811 * | 0.000 |
| | Amino acid | 0.0000032811 * | 0.000 |
| | Stearic acid | 0.0000032179 * | 0.000 |
| | Benzene | 0.0000031829 * | 0.000 |
| | Toulene | 0.0000030693 * | 0.000 |
| | Cetyl Palmitate | 0.0000032873 * | 0.000 |
| | Methylene glycol | 0.0000030573 * | 0.000 |
| | Sodium chloride | 0.0000032563 * | 0.000 |
| | Potassium cetyl sulphate | 0.0000032689 * | 0.000 |
| | Tea Tree Oil | 0.0000034594 * | 0.000 |

* Mean value of five replicates $\pm$ standard deviation ($n$ = 5).

### 3.2. Accuracy Test

Spiking is the most common technique to ensure consistency since it describes sample effects and the specific raw sample that should always get certified reference material. In other words, the raw samples were prepared with an absolute concentration of the standard sample (known concentration) to assess the efficiency of the modified electrode. Special precautions must be taken when spiking is done in the field by non-laboratory staff. It is recommended that all spike preparation procedures were carried out by a trainer in a laboratory so that the field activity consists solely of the samples of a prepared spiking solution and to ensure that spikes are added correctly and reproducibly. The field spikes were duplicated to record the reproducibility of the procedure.

The developed method's recovery rate was 96.6–97.5% with an acceptable relative standard deviation (RSD) less than 1%, as shown in Table 4. The amount of mercury in the cosmetic product samples tested complies with the standard requirement. Thus, the developed method demonstrated the potential application for mercury determination in cosmetics. The sensor has remarkable advantages, such as higher sensitivity, wider linear range, and a lower limit for detection than other sensors (Table 5).

**Table 4.** Recovery Study of the Developed Sensor, as Reported by Bohari et al. [42].

| | Added (ppm) | Found (ppm)Mean $\pm$ STD | Recovery (%) | RSD (%) |
|---|---|---|---|---|
| Developed sensor | 0.03 ppm | 0.03 $\pm$ 0.38 | 96.6% | 0.43% |
| | 6 ppm | 5.7 $\pm$ 0.45 | 97.5% | 0.52% |
| | 10 ppm | 9.49 $\pm$ 0.43 | 97.3% | 0.64% |

**Table 5.** A comparison of different types of sensors for mercury detection.

| References | Real Samples | Type of Sensor | LOD | Response Time |
|---|---|---|---|---|
| This study | Mercury | Electrochemical sensor | 0.08 ppm | 70 s |
| [25] | Fish tissue, natural surface water, drinking water, and seawater. | Colorimetric | 0.5 ppb | 1–90 min |
| [30] | Human saliva | SERS sensor | 2.3 ppt | - |
| [54] | Water | Fluorescent | 1.02 ppb | |
| [55] | Aqueous solution | UV-vis optical sensor | 1.4 ppb | - |
| [56] | Human serum, water, and milk. | Fluorescent sensors | 1.33 ppt | 10 min |
| [57] | Biological and environmental systems | Electrochemical sensor | 7 ppt | 500 s |
| [58] | Hg(II) in water | Electrochemical sensor | 0.06 $\mu$M | 90 s |
| [59] | Trace Hg(II) in different real samples with | Electrochemical sensor | 0.03 $\mu$g/L (0.15 nM) | 300 s |
| [60] | Mercury in water using | Electrochemical sensor | 1.0 ppb | 600 s |
| [61] | Water bodies | Electrochemical sensor | - | 100 s |
| [62] | Water | Electrochemical sensor | 0.3 $\mu$M | 120 s |
| [63] | Water | Electrochemical sensor | 0.208 $\mu$M | 150 s |
| [64] | Water | Electrochemical sensor | 0.017 $\mu$M | 150 s |
| [34] | Mercury (II) ions in water | Electrochemical sensor | 0.74 ppb | 2 min |
| [65] | Fish and seawater samples | Electrochemical sensor | $1.35 \times 10^{-8}$ mol/L | 60 s |
| [66] | Water and biological samples | Electrochemical sensor | 6.6 $\mu$M | NA |
| [67] | Tap water | Electrochemical sensor | 8.43 $\mu$M | 1 min |
| [35] | Environment water | Electrochemical sensor | (1.02 nM) | 2 hours |
| [68] | Amazon river | Electrochemical sensor | $5-300$ $\mu$g/L | 60 s or 300 s (without stirring) |
| [69] | Water | Electrochemical sensor | 0.017 $\mu$g/L | 500 s |

## 4. Conclusions

In this study, the modified electrode's electrochemical behaviour is developed based on PANI, MWCNTs, AuNPs, and ITO to determine mercury in cosmetic products. Combining the advantages of these nanocomposites and nanomaterials (PANI/MWCNTs/AuNPs) in MB presence are effectively enhanced electron transfer rate and promoted the increase in current response for mercury detection. The developed electrochemical sensor was performed under optimum conditions that found high precision, repeatability, high reproducibility & selectivity, and high sensitivity towards mercury using Tukey Post Hoc, $p < 0.05$. Compared to those existing methods, electrochemical sensor methods for mercury determination have various advantages such as simplicity, low cost, rapidity, effectiveness,

and high sensitivity. The electrochemical approaches in this study are indicated to monitor harmful mercury in cosmetic products on-site.

**Supplementary Materials:** The following are available online at https://www.mdpi.com/2079-9 284/8/1/17/s1, Figure S1. Computer-controlled potentiostat/galvanostat with a standard three-electrode system connected to an electrochemical cell where Tris-HCl and mercury used as analytes.

**Author Contributions:** Conceptualisation, S.S. (Shafiquzzaman Siddiquee), S.S. (Suryani Saallah), M.M. and S.E.A.; investigation, N.A.B.; resources, S.S. (Shafiquzzaman Siddiquee); writing—original draft preparation, N.A.B.; writing—review and editing, S.S. (Shafiquzzaman Siddiquee) and S.S. (Suryani Saallah); supervision, S.S. (Shafiquzzaman Siddiquee), M.M. and S.E.A. All authors have read and agreed to the published version of the manuscript.

**Funding:** This research was funded by University Malaysia Sabah Scheme, UMS Great: Development of nanocomposite nanofiber based on Indium Tin Oxide (ITO) for mercury detection in cosmetic products (GUG0112-1/2017).

**Data Availability Statement:** Not applicable.

**Conflicts of Interest:** The authors declare no conflict of interest.

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
