# Peer review of "Electrochemical Behaviour of Real-Time Sensor for Determination Mercury in Cosmetic Products Based on PANI/MWCNTs/AuNPs/ITO"

_cosmetics, doi:10.3390/cosmetics8010017_

Round 1

Reviewer 1 Report

The manuscript deals with the interesting topic. The study is well-designed and manuscript is well-written. I have some minor remarks: 

- Introduction should contain more data on Hg toxicity

-Figure legends should be more informative

-Conclusion should be more general in terms of the results of the study (no need to state the exact levels of LOD, etc.)

Author Response

REVIEWER 1

NO.

COMMENT

RESPONSE

REVISION

1

Introduction should contain more data on Hg toxicity

We sincerely appreciate all the valuable comments and suggestions from the reviewer, which helped us improve our manuscript’s quality.

Line 68-124

2

Figure legends should be more informative

The sentence has been revised and corrected.

Figure 1. The schematic diagram of a biosensor incorporates a living organism or product derived from a living system as the recognition element or a bioreceptor and a transducer to convert a biological reaction into a measurable signal or indication.

Figure 2. DPV on the different mercury concentrations with PANI/MWCNTS/AuNPs/ITO (pH 7.0 Tris–HCl, 1 mM MB). Inset: Calibration curve between the oxidation peak currents obtained by DPV and mercury concentrations.

Figure 3. The cyclic voltammogram and the bar chart of a) multiple cycling, b) reproducibility, and c) repeatability test of the modified ITO in the presence of standard mercury, in 50 mM Tris-HCl, pH 7.0, 1 mM MB, n=5.

Figure 4. Bar chart of the developed sensor's storage stability in the presence of standard mercury, in 50 mM Tris-HCl, pH 7.0, 1 mM MB, n=5.

3

Conclusion should be more general in terms of the results of the study (no need to state the exact levels of LOD, etc.)

The sentence has been revised and the unnecessary sentences were deleted.

n this study, the modified electrode's electrochemical behaviour is developed based on PANI, MWCNTs, AuNPs, and ITO to determine mercury. Combining the advantages of these nanocomposites and nanomaterials (PANI/MWCNTs/AuNPs) in MB's presence are effectively enhanced electron transfer rate and promoted the increase in current response for mercury detection. The developed electrochemical sensor was performed under optimum conditions that found high precision, repeatability, high reproducibility & selectivity, and high sensitivity towards mercury using Tukey Post Hoc, p < 0.05. Compared to those existing methods, electrochemical sensor methods for mercury determination have various advantages such as simplicity, low cost, rapidity, effectiveness, and high sensitivity. The electrochemical approaches in this study are indicated to monitor harmful mercury in cosmetic products on-site.

Reviewer 2 Report

In the Manuscript (cosmetics-1047754), a novel simple method is proposed for a fast and simple procedure technique for analysing the mercury levels in cosmetic products. According to this reviewer, the manuscript is well written and the experiments well planned out and systematic to prove the properties that are claimed. A few suggestions prior to accepting the manuscript are as follows

  1. The manuscript has many grammatical errors that needs to be fixed through proof reading.
  2. Can the authors fix the statement

“suggesting that the sensor 306 exhibited high mercury selectivity towards other compounds “. Right now it is confusing whether selectivity is toward mercury or other compounds.

  1. The authors need to discuss other studies/methods that deal with detecting mercury ions in different media and compare them with their own results in the introduction and results sections, respectively. Some such studies include

Environmental Science & Technology 49 (3), 1578-1584 and Chemistry. 2017 Nov 16; 23(64): 16219–16230. doi: 10.1002/chem.201702871 and Angew. Chem., Int. Ed. 2007, 46, 4093– 4096 and Microchim Acta 2012, 177, 341– 348 ; just to name a few.

  1. The authors claim a detection range of 0.01-10.00 ppm but a a limit of detection of 0.03 ppm. Please clarify. Is it a typo?
  2. The authors claim sensitivity of 0.03 ppm. This is not the units for sensitivity. Could you please fix this error?
  3. Are the authors suggesting that the sensor is only reproducible over 14 days? Can this be improved in future do you think?
  4. Can the selectivity tests be presented with the mercury mean difference value too so the readers can compare with the other species? The selectivity is not clear in its current form of presentation. Or perhaps further explanation can be provided?

Author Response

REVIEWER 2

COMMENT

RESPONSE

REVISION

The manuscript has many grammatical errors that needs to be fixed through proof reading.

We sincerely appreciate all the valuable comments and suggestions from the reviewer, which helped us improve our manuscript’s quality.

The whole manuscript has revised accordingly your comments

Can the authors fix the statement

“suggesting that the sensor 306 exhibited high mercury selectivity towards other compounds “. Right now it is confusing whether selectivity is toward mercury or other compounds.

The sentence has been revised and corrected.

The results showed that the current mercury responses were significantly higher than other substances, suggesting that the sensor exhibited high mercury selectivity towards mercury than other compounds.

The authors need to discuss other studies/methods that deal with detecting mercury ions in different media and compare them with their own results in the introduction and results sections, respectively. Some such studies include

Environmental Science & Technology 49 (3), 1578-1584 and Chemistry. 2017 Nov 16; 23(64): 16219–16230. doi: 10.1002/chem.201702871 and Angew. Chem., Int. Ed. 2007, 46, 4093– 4096 and Microchim Acta 2012, 177, 341– 348 ; just to name a few.

We sincerely thank you for the suggestion of references.

Esmaielzadeh Kandjani, A., Sabri, Y.M., Mohammad-Taheri, M., Bansal, V. and Bhargava, S.K., 2015. Detect, remove and reuse: a new paradigm in sensing and removal of Hg (II) from wastewater via SERS-active ZnO/Ag nanoarrays. Environmental Science & Technology, 49(3), pp.1578-1584.

Worthington, M.J., Kucera, R.L., Albuquerque, I.S., Gibson, C.T., Sibley, A., Slattery, A.D., Campbell, J.A., Alboaiji, S.F., Muller, K.A., Young, J. and Adamson, N., 2017. Laying waste to mercury: inexpensive sorbents made from sulfur and recycled cooking oils. Chemistry (Weinheim an der Bergstrasse, Germany), 23(64), p.16219.

Lee, J.S., Han, M.S. and Mirkin, C.A., 2007. Colorimetric detection of mercuric ion (Hg2+) in aqueous media using DNA‐functionalized gold nanoparticles. Angewandte Chemie International Edition, 46(22), pp.4093-4096.

Example of suggested references are included on the Introduction and results sections

The authors claim a detection range of 0.01-10.00 ppm but a limit of detection of 0.03 ppm. Please clarify. Is it a typo?

Absolutely apology for typing error.

Currently actual data included in abstract and Limit of Detection (LOD)

Are the authors suggesting that the sensor is only reproducible over 14 days? Can this be improved in future do you think?

According to Mathias [1], the recovery percentage should be above 80% to represent high reproducible. Hence, 14th day is still acceptable reproducible (95%) of its original value. It can be improved in future by using additive or advanced nanotechnology; materials can effectively be made stronger, lighter, more durable, more reactive, more sieve-like, or better electrical conductors, among many other traits.

Reviewer 3 Report

In this study, the modified electrode's electrochemical behavior is developed based on PANI,

Bohari et al. developed sensing platform (method) for inorganic mercury determination in terms of safety of cosmetic products but also all other products on the market that contain inorganic mercury. It is interesting combination of nanocomposites and nanomaterials which in the presence of MBs enhance electron transfer rate and increase response for mercury determination. All determination parameters are calculated and imply that method could be used for mercury determination by sensing platform. Idea is good and applicable. 

Authors provided also details when different excipience are used.  Interesting idea and very applicable approach. Well done. 

Please just check once again typing errors and English. 

Author Response

REVIEWER 3

NO.

COMMENT

RESPONSE

REVISION

1

Please just check once again typing errors and English. 

We sincerely appreciate all the valuable comments and suggestions from the reviewer, which helped us improve our manuscript’s quality.

The grammatical error has been corrected.